# Green Synthesized Metal Oxide Nanoparticles Mediate Growth Regulation and Physiology of Crop Plants under Drought Stress

**DOI:** 10.3390/plants10081730

**Published:** 2021-08-21

**Authors:** Nadiyah M. Alabdallah, Md. Mahadi Hasan, Inès Hammami, Azzah Ibrahim Alghamdi, Dikhnah Alshehri, Hanan Ali Alatawi

**Affiliations:** 1Department of Biology, College of Science, Imam Abdulrahman Bin Faisal University, P.O. Box 1982, Dammam 31441, Saudi Arabia; nmalabdallah@iau.edu.sa (N.M.A.); ihammami@iau.edu.sa (I.H.); azalghamdi@iau.edu.sa (A.I.A.); 2State Key Laboratory of Grassland Agro-Ecosystems, School of Life Sciences, Lanzhou University, Lanzhou 730000, China; 3Department of Biological Sciences, College of Science, University of Tabuk, Tabuk 74191, Saudi Arabia; dalshehri@ut.edu.sa (D.A.); halatwi@ut.edu.sa (H.A.A.)

**Keywords:** nanoparticles, abiotic stress, hydrogen peroxide, malonaldehyde, oxidative stress

## Abstract

Metal oxide nanoparticles (MONPs) are regarded as critical tools for overcoming ongoing and prospective crop productivity challenges. MONPs with distinct physiochemical characteristics boost crop production and resistance to abiotic stresses such as drought. They have recently been used to improve plant growth, physiology, and yield of a variety of crops grown in drought-stressed settings. Additionally, they mitigate drought-induced reactive oxygen species (ROS) through the aggregation of osmolytes, which results in enhanced osmotic adaptation and crop water balance. These roles of MONPs are based on their physicochemical and biological features, foliar application method, and the applied MONPs concentrations. In this review, we focused on three important metal oxide nanoparticles that are widely used in agriculture: titanium dioxide (TiO_2_), zinc oxide (ZnO), and iron oxide (Fe_3_O_4_). The impacts of various MONPs forms, features, and dosages on plant growth and development under drought stress are summarized and discussed. Overall, this review will contribute to our present understanding of MONPs’ effects on plants in alleviating drought stress in crop plants.

## 1. Introduction

By 2050, the world population is expected to reach nearly 9.6 billion people, requiring a 70–100% increase in agricultural productivity to fulfill the world’s food needs [1,2]. However, decreasing fertile area, water scarcity, the effects of global warming, and the low efficacy of present fertilizers and pesticides exacerbate abiotic stresses on crops, lowering their yields. Drought, for example, costs billions of dollars in crop yield loss each year [3,4]. As a result, decreasing food production is a serious concern. Drought-tolerant crop varieties have taken a long time to develop, yet there are still few economically feasible vigorous drought-tolerant species [3,4,5,6]. Simultaneously, public anxiety about the safety of the transgenic crop is high [7]. Thus, innovative technologies that protect the plants from drought stress are required to ensure food security in a safe and sustainable manner.

Nanotechnology has been commonly applied in the food, medical, and agricultural industries throughout the world [8]. Numerous metallic nanoparticles (MONPs), such as titanium dioxide (TiO_2_), iron oxide (Fe_3_O_4_), and zinc oxide (ZnO), have gained considerable attention in recent years due to their environmentally favorable use in agriculture. MONPs can be synthesized in a variety of ways, including green, chemical, and physical processes. However, green synthesis of MONPs has been extensively utilized in the agricultural sector [9]. MONPs have been shown to have positive impacts on crop growth in recent years. These effects varied according to the form, origin, and size of the MONPs, as well as the plant species and the time of MONPs exposure to crops [10,11].

Recently, MONPs have been used to increase plant tolerance to harsh environments. MONPs have been utilized to protect plants from oxidative stress by increasing the activity of antioxidant enzymes such as superoxide dismutase (SOD), catalase (CAT), and peroxidase (POX) [12]. MONPs have the potential to reduce the detrimental impact of drought on plant physiological functions by lowering malondialdehyde (MDA) and hydrogen peroxide (H_2_O_2_) contents and maintaining photosynthetic systems [12,13]. Under drought stress, they play a role in signaling pathways, defense, metabolism, and regulatory activities. For example, TiO_2_ nanoparticles (TiO_2_NPs) decreased oxidative damage and lipid peroxidation in response to drought stress, as shown by reduced H_2_O_2_ and MDA concentrations [14]. MONPs can penetrate chloroplasts and react with the photosystem II reaction center, hence increasing electron transmission, oxygen evolution, and light absorption in chloroplasts under drought-induced oxidative stress [15]. Despite their commercial significance and prevalence in a variety of commercial products, there is obviously a rising public concern about the toxicological and environmental impacts of MONPs [16]. Excessive MONPs caused physiological abnormalities and oxidative stress in crops, resulting in a decrease in gas exchange characteristics and antioxidant enzyme activities [17,18,19,20]. Numerous studies found that MONPs reduced the mitotic index and disrupted cell division phases in the root tips and altered the gene expressions associated with root growth [21,22,23]. MONPs cause indirect toxicities by altering the growth medium and soil bacterial communities, also causing co-contaminants to be absorbed by plants [24,25]. MONPs could increase or decrease crop growth and yield, and they can be transferred into the food chain with unknown consequences to humans and animals [17,20,26,27]. Therefore, MONPs may not be widely used in agriculture.

There have been a number of research studies on the synthesis and characterization of MONPs, as well as their role in abiotic stress tolerance, but only a few reports have been published that summarize the green synthesis of MONPs and drought stress tolerance in plants. Furthermore, the current study highlights recent improvements in the use of MONPs itself, whether given directly through hydroponics, or through the soil, to boost plant growth and drought stress tolerance in a variety of plant environments.

## 2. Synthesis and Characterization of Metal Oxide Nanoparticle

The conventional methods for producing MONPs are based on physical and chemical processes that involve the use of dangerous and expensive substances, which require a large amount of energy and have a detrimental impact on the environment [9]. The green synthesis of MONPs has received a lot of attention recently since it is an innovative method for developing engineered materials [28]. In comparison to conventional chemical and/or physical processes, green synthesis of MONPs by various organisms (algae, fungi, bacteria, plants, etc.) provides a dependable, limited, and environmentally sustainable option [10,11,28]. During biosynthesis, the green production of MONPs results in the development of capped nanostructures with proteins/biomolecules from the organisms. Such capping agents inhibit the aggregation of nanoparticles and play a significant role in the nanosystem’s stabilization [11,29]. Green synthesis of MONPs is illustrated schematically in Figure 1.

The green synthesis of MONPs can be done using simple and cost-effective methods that do not pollute the environment [28,29]. TiO_2_ nanoparticles (TiO_2_NPs) offer a wide range of uses in the environmental, industrial, and medicinal sectors [30]. The non-toxic TiO_2_NPs exhibit strong oxidation potential, show considerable photo-catalytic activity, and have unusual optical and chemical stability. Additionally, they have antimicrobial and antibacterial catalytic properties, which enable them to be used in a variety of industrial applications, including photocatalysts, catalyst supports, and pigments [31,32]. TiO_2_ exhibited improved biocompatibility and stability, most likely as a result of the capping agent coated on the surface [31]. Green synthesized TiO_2_NPs have been synthesized mostly using fungi, bacteria, and plants (Table 1). In general, X-ray diffraction (XRD), atomic force microscopy (AFM), thermogravimetric analysis (TGA), Fourier transform infrared (FTIR) spectroscopy, and transmission electron microscopy (TEM) have been used to characterize TiO_2_NPs.

ZnO nanoparticles (ZnONPs) have been extensively employed in the formulation of sunscreen lotions and cosmetics and are also used as biocidal agents/disinfectants due to their UV absorption capacity and excellent photostability [61,62,63]. Additionally, they show antibacterial and anticancer properties [64,65]. ZnONPs (16–108 nm) with antibacterial activity were synthesized using plant *Parthenium hysterophorous* [54]. Although ZnONPs are stable and affordable to synthesize, aggregation of chemically derived NPs can cause their instability and expansion in size due to their high surface energy. Capping with modifying agents or surfactants such as polyethylene glycol (PEG), polyethylene oxide (PEO), and polyvinyl pyrrolidone (PVP) results in their significant size reduction contributing to the stability of nanoparticles [66,67,68]. Iron oxide nanoparticles (Fe_3_O_4_NPs) have prospective applications in a variety of biomedical fields, including delivery of drug, cancer diagnosis, treatment, and the imaging of nuclear magnetic resonance [69,70]. Apart from the conventional chemical approaches, there is a growing trend in the utilization of green methods to synthesize Fe_3_O_4_NPs. To limit their growth of the NPs, polymers, organic capping agents, or structural hosts are utilized. Phenolic compounds act as capping agents, improving colloidal solution stability and preventing nanoparticle aggregation. One of the non-toxic, naturally occurring polyphenolic substances derived from plants is tannins. Herrera-Becerra et al. [71] used tannins to create green synthesized magnetic hematite (Fe_2_O_3_) nanoparticles with a diameter of only about 10 nm and a pH of 10. Using the *Plantago* spp. peel extract of *Malus domestica* as a capping agent, Venkateswarlu et al. [58] were able to synthesis spherical Fe_3_O_4_NPs with an average diameter of 50 nm, while aqueous leaf extract of *Tridax procumbens* was used to make capped Fe_3_O_4_ with a diameter of 80–100 nm [59]. Comprehensive surface characterization approaches such as surface characteristics, chemical properties, and spatial patterns of functional groups are utilized to gain a deeper understanding of surface properties [72]. Fe_3_O_4_NPs are investigated using a variety of fundamental techniques, including FTIR spectroscopy, XRD, scanning electron microscopy (SEM), TEM, and TGA analysis [55,56,57,58].

## 3. Mode of Action of Metal Oxide (MONPs) Nanoparticles under Drought Stress

Drought is a common abiotic source of stress that drastically reduces crop yield in arid environments [6,73,74]. Water is necessary for plant viability and nutrient transport. The viability of plants is harmed by water shortages or drought [4,75]. The use of various MONPs can be used to alleviate water scarcity (Figure 2).

MONPs, which are detailed under the section heading below, have been shown to improve plant drought stress tolerance. 

### 3.1. TiO_2_NPs Nanoparticles Mediated Drought Stress Tolerance

TiO_2_NPs are one of the most frequently utilized nanoparticles, with applications in cosmetics and skincare, antibacterial air-cleaning goods, and wastewater decomposition [76,77]. Due to the photocatalytic capabilities, the majority of studies using TiO_2_NPs at the foliar level have shown a beneficial effect on plants. According to Jaberzadeh et al. [78], exposure to low concentrations of TiO_2_NPs could significantly reduce the detrimental impact of drought in wheat. Under drought stress conditions, TiO_2_NPs increased plant height, ear weight, ear and seed number, yield, biomass, and harvest index [78]. In addition, TiO_2_NPs enhanced substantially gluten and starch content under drought stress [79]. 

The exogenous application of TiO_2_NPs resulted in an increase in wheat shoot fresh and dry weight, as well as an increase in photosynthetic pigments in wheat [80] and *Linum usitatissimum* [81] under drought stress. Activating photosynthesis and nitrogen metabolism may boost *Triticum aestivum* plant growth. TiO_2_NPs is a form of photocatalyst that can hydrolyze light into oxygen, electrons, and protons. The generated electron and proton are then transferred to a plant’s electron transfer chain during the light reaction stage, thereby increasing the rate of photosynthesis [81]. The enhancement of secondary metabolites like phenolic compounds by MONPs has been recognized as a strategy for alleviating abiotic stress. TiO_2_NPs had a considerable impact on secondary metabolites in a drought environment; namely, when *Lallemantia iberica* was subjected to moderate drought stress, TiO_2_NPs caused a considerable rise in phenolic compounds as well as total flavonoid content [82]. It has been reported that TiO_2_NPs reduce the H_2_O_2_ and MDA contents in *Triticum aestivum* [83] (Table 2). 

Antioxidant activity of enzymes such as CAT and APX were greatly elevated in plants treated with TiO_2_NPs under drought stress, showing that the defensive strategy has been activated by the plants [83].

### 3.2. ZnONPs Mediated Drought Stress Tolerance

Zn influences the structure, function, and performance of a wide range of enzymes [87]. There is also substantial proof that ZnONPs boost crop production and biomass accumulation when plants are subjected to drought stress. For instance, Dhoke et al. [92] examined the influence of ZnONPs on the growth of *Vigna radiata* seedlings and found that the application of the ZnONPs boosted the root biomass and above-ground tissues. Photosynthesis has an impact on plant growth, productivity, and drought tolerance, and it is regarded to be the foundation of life on Earth. Taken together with stomatal conductance (*g_s_*), it is the most important step in the production of crop yield [93]. ZnONPs were found to have a beneficial effect under drought stress. These nanoparticles increased photosynthetic activity, chlorophyll content, transpiration rate, stomatal conductance, and water use efficiency in maize seedlings [89]. ZnONPs assisted in the stabilization of the chloroplast and mitochondrial ultrastructures of water-stressed *Zea mays*, hence increasing photosynthetic efficiency [89]. This could be due to the osmolyte accumulation such as proline and sugars required for the osmotic adjustment function [89]. Additionally, it may contribute to the maintenance of cell membrane integrity and the increase in relative water content (RWC), which may represent plant metabolic functions. Thus, the authors proposed a nanotechnology-based technique for increasing plant growth and yield. Dimkpa et al. [87] found that ZnONPs can hasten *Sorghum bicolor* growth, increase yield, enrich edible grains with key elements such as zinc, and improve nitrogen uptake during drought stress conditions. ZnONPs increased grain nitrogen translocation by 84% compared to the drought control and recovered total N levels. In addition, ZnONPs application to drought-affected seedlings increased overall K uptake (16–30%) and grain K uptake (123%) in comparison to the drought control [87]. Foroutan et al. [90] showed that ZnONPs treatment could significantly increase drought tolerance in distinct *M. peregrina* species during water deficit conditions by increasing the antioxidant polyphenol oxidase (PPO) and peroxidase (POD) activities and osmoprotectant content (Table 2). In comparison to the control, foliar application of ZnONPs reduced oxidative stress and increased leaf SOD and POD activities [88]. Increased antioxidant enzyme activity and decreased oxidative stress indicators in wheat leaves may represent a stress tolerance mechanism under stressful conditions [88]. SOD assists in the detoxification of superoxide (O_2_^−^) under drought-induced oxidative stress by activating dismutation reaction and converting it to O_2_ and H_2_O_2_. Finally, these antioxidant enzymes act harmoniously to prevent the generation of damaging ROS. 

### 3.3. Fe_3_O_4_NPs-Mediated Drought-Stress Tolerance

A variety of physiological processes in plants, such as the synthesis of chlorophyll content, photosynthetic activity, and metabolism, are influenced by iron levels in the environment [91]. Numerous findings suggest that Fe-based NPs promote plant development in non-stress conditions. Fe_3_O_4_NPs have beneficial impacts on plant development even at relatively low doses. Iron nanoparticles application may be an effective technique for increasing iron absorption through the roots of plants and increasing their stability during drought stress. Alidoust and Isoda [94] conducted research on the impact of Fe_3_O_4_NPs on *Glycine max*. A foliar application of Fe_2_O_3_NP coated with citric acid resulted in considerable increases in root length and photosynthetic rate [92]. *Fragaria × ananassa* plantlets treated with Fe_3_O_4_NPs were more effective than untreated plantlets in dealing with drought stress conditions. Mozafari et al. [91] showed that Fe_3_O_4_NPs with sizes ranging from 40 to 53 nm considerably improved the plant growth, relative water content, and photosynthetic pigments of a *Fragaria × ananassa* under drought conditions. Additionally, Fe_3_O_4_NPs application increased the *Fragaria × ananassa* membrane stability index, resulting in higher activities of SOD and POD enzymes and a lower quantity of H_2_O_2_. By increasing the effectiveness of redox processes and/or activating H_2_O_2_-metabolizing enzymes, Fe_3_O_4_NPs may be used to reduce or eliminate H_2_O_2_ production. Fe_3_O_4_NPs-treated *Oryza sativa* plants experienced a rise in their biomass, antioxidant enzyme activities, photosynthetic efficiency, and nutrient uptake during drought stress [12]. These findings suggest that increasing the amount of iron applied to plants in the form of nanoparticles could be beneficial to their growth and physiology. 

## 4. Conclusions and Future Perspective

Drought stress is responsible for the majority of crop production decreases globally. MONPs must be given major study priority due to their capability to promote drought stress tolerance in agricultural crops. The purpose of this review was to discuss the use of MONPs to boost plant development in drought-stressed conditions, as well as their potential application in agricultural production. A significant step in the application of nanotechnology in sustainable farming will be a move from testing/using MONPs in plants to creating MONPs based on agricultural demands. Nevertheless, MONPs mobility and environmental impact should be extensively studied to ensure their safe usage in agricultural production. To further comprehend how the MONPs increased plant productivity and drought stress tolerance, the elements such as the size and concentration of these nanoparticles and the cultivation technique must be all specified or explained.

## Figures and Tables

**Figure 1 plants-10-01730-f001:**
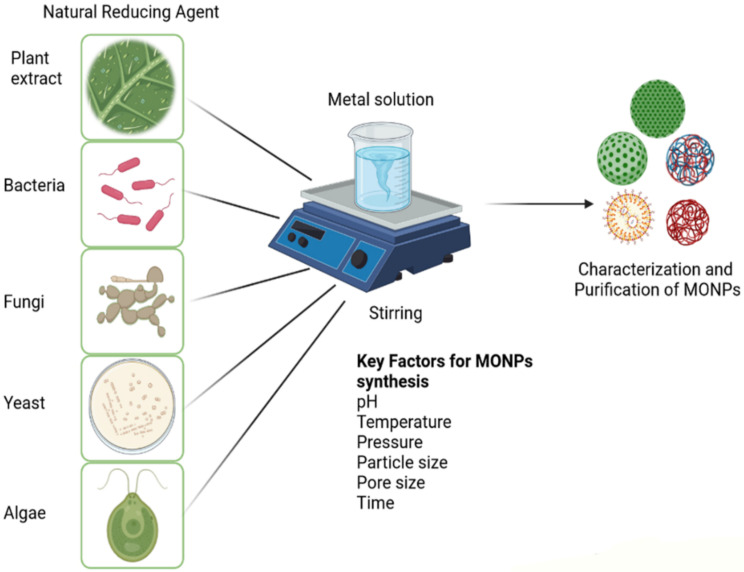
The green synthesis of metal oxide nanoparticles (MONPs) is represented schematically. Created with Biorender.

**Figure 2 plants-10-01730-f002:**
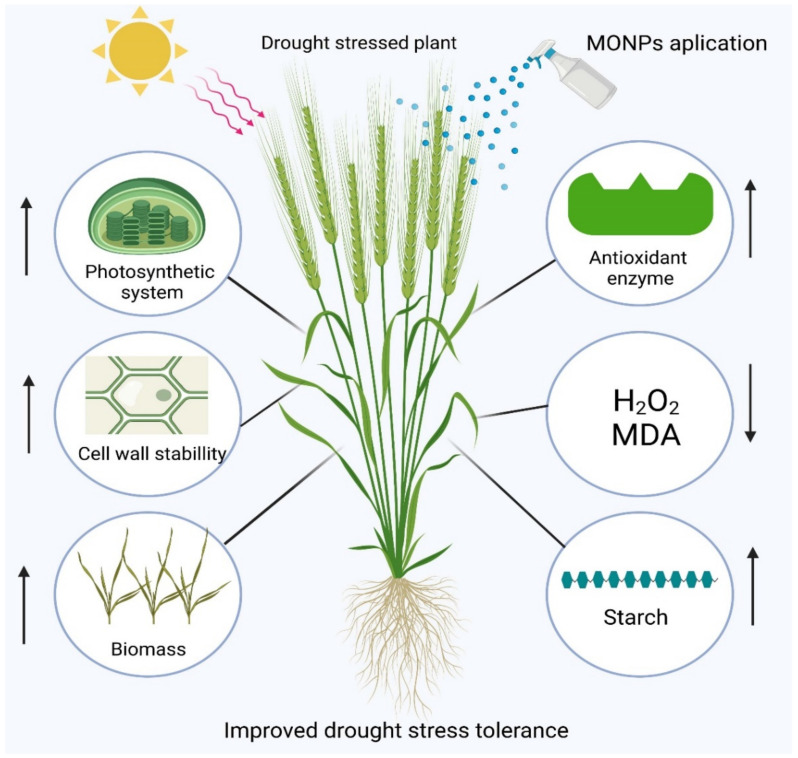
Metal oxide nanoparticles (MONPs) induced drought stress tolerance in plants through a general mechanism. Created with Biorender.

**Table 1 plants-10-01730-t001:** Metal oxide nanoparticles (MONPs) synthesized from different biological substrates.

Metal Oxide Nanoparticles (MONPs)	Biological Substrate	Name of Source	Size (nm)	Shape	References
**TiO_2_**	Fungi	*Aspergillus flavus*	62–74	spherical/oval	[33]
	Bacteria	*Aeromonas hydrophila*	28–54	Spherical	[34]
		*Bacillus mycoides*	40–60	polymorphic	[35]
		*Lactobacillus* sp.	10–70	Spherical	[36]
	Plants	*Cicer arietinum* L.	14	Spherical	[37]
		*Citrus sinensis* L.	19	Tetragonal	[38]
		*Annona squamosa* L.	23	Polydisperse	[39]
		*Ocimum basilicum* L.	50	Hexagonal	[40]
		*Solanum trilobatum* L.	70	Spherical	[41]
		*Jatropha curcas* L.	25–100	Spherical	[42]
		*Moringa oleifera* Lam.	100	Spherical	[43]
**ZnO**	Algae	*Sargassum murticum*	30–57	Spherical	[44]
	Bacteria	*Lactobacillus sporoge*	5–15	Hexagonal	[45]
		*Staphylococcus aureus*	10–15	Acicular	[46]
		*Acinetobacter schindleri*	20–100	Spherical	[47]
	Plants	*Azadirachta indica*	18	Spherical	[48]
		*Citrus paradise*	19	Polyhedron	[49]
		*Solanum nigrum*	30	Hexagonal	[50]
		*Aloe barbadensis*	25–40	Spherical	[51]
		*Vitex negundo*	75–80	Spherical	[52]
		*Lycopersicon esculentum*	40–100	Spherical	[53]
		*Parthenium hysterophorous*	16–108	Spherical	[54]
**Fe_3_O_4_**	Bacteria	*Klebsiella oxytoca*	2–5	Spherical	[55]
		*Actinobacter* spp.	100	Spherical	[56]
	Plants	*Vitis vinifera*	30	-	[57]
		*Plantago* spp.	>50	Spherical	[58]
		*Tridax procumbens*	80–100	Irregular spheres	[59]
		*Punica granatum*	100–200	-	[60]

**Table 2 plants-10-01730-t002:** Effects of application of metal oxide nanoparticles (MONPs) on drought stress in crop plants.

Metal Oxide Nanoparticles (MONPs)	Plant Species	Concentration of Applied Metal Oxide Nanoparticles	Drought Level	Effects	Outcome	References
**TiO_2_**	Wheat (*Triticum aestivum* L. cv. Pishtaz)	2.1, 4.3, and 6.6 mmol/L	Withheld water	Increased plant height, ear weight, ear number, seed number, final yield, biomass, harvest index, and starch contents	Increased drought stress tolerance	[78]
	*Lallemantia iberica*	6.6 mmol/L	75% and 35% of Field Capacity (FC)	Significant increase in phenolic content and total flavonoid and antioxidant activity	Alleviated drought stress	[82]
	Wheat (*Triticum aestivum* L. cv. Pishgam)	0, 10.9, 21.7, and 43.4 mmol/L	PEG—induced drought stress (−0.4 and −0.8 MPa)	Increased germination percentage, germination energy, germination rate, root length, shoot length, root fresh weight, shoot fresh weight, and vigor index	Decreased negative effects of drought stress on wheat plants	[80]
	Basil (*Ocimum basilicum* L.)	2.1 and 6.6 mmol/L	Field capacity (FC)—40%	Improved relative water content, catalase activity, and anthocyanin content	Enhanced drought tolerance in basil plants	[84]
	Wheat (*Triticum aestivum* L. cv. Pishgam)	10.8, 21.7, and 43.7 mmol/L	Field capacity (FC)—75% and 50%	Enhanced relative water content (RWC), enhanced total chlorophyll, carotenoids, stomatal conductance, transpiration, CAT activity, APX activity. Significantly reduced H_2_O_2_ and MDA content	Protected oxidative damage from drought stress	[83]
	Cotton (*Gossypium barbadense* L.)	0.5, 1.0, 2.1 and 4.3 mmol/L	Withheld water	Increased total phenolics, soluble proteins, free amino acids, proline content, and antioxidant capacity	Increased drought tolerance in cotton plants	[85]
	*Linum usitatissimum* cv. Olajonzon	0, 0.2, 2.1, and 10.8 mmol/L	Field capacity (FC)—50%	Enhanced chlorophyll and carotenoids contents. Decreased MDA and H_2_O_2_ content	Prevented oxidative injury and increased drought tolerance	[81]
**ZnO**	*Solanum melongena* L. cv. Soma	0, 1.0, and 2.1 mmol/L	60% of crop evapotranspiration (ETc)	Increased membrane stability index (MSI), relative water content (RWC), and photosynthetic efficiency	Improved drought-tolerant cultivar	[86]
	*Sorghum bicolor* var. 251	0.02, 0.06, and 0.1 mmol/L	Field capacity (FC)—40%	Improved grain (22–183%) yield, improved (84%) grain N translocation	Increased drought stress tolerance	[87]
	(*Triticum aestivum* var. *Lassani—2008*)	(0, 0.5, 1.0, 2.1 mmol/L)	Field capacity (FC)—70% and 35%	Increased leaf chlorophyll contents, SOD, and POD activities	Higher drought tolerance in a wheat variety	[88]
	*Zea mays* L. cv. Jidan 27	2.1 mmol/L	Field capacity (FC)—45%	Increased photosynthetic pigment, photosynthetic rate, water use efficiency, UDP-glucose pyrophosphorylase, phosphoglucoisomerase, and cytoplasmic invertase	Alleviated drought stress by increasing photosynthetic capacity	[89]
**Fe_3_O_4_**	*Moringa peregrina* (Forssk.)	10.8 and 21.1 mmol/L	Field capacity (FC)—50%	Enhanced POD and PPO activities	Mitigated drought stress by increasing antioxidant activity	[90]
	Strawberry (*Fragaria × ananassa* Duch.)	40–53 nanometer size	0, 5, and 10%) of polyethylene glycol (PEG 6000)	Increased pigment levels, relative water content, membrane-stability index and decreased MDA and H_2_O_2_ content	Improved drought tolerance by alleviating oxidative injury	[91]
	*Oryza sativa* cv. Super Basmati Rice	Combined application oxide and hydrogel nanoparticles (0.5, 1.08, 2.1 mmol/L)	Field capacity (FC)—35%	Increased biomass, antioxidant enzyme activities, photosynthesis efficiency, nutrient acquisition	Improved drought tolerance	[12]

## Data Availability

Not applicable.

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
