# Peer review of "Green Synthesized Metal Oxide Nanoparticles Mediate Growth Regulation and Physiology of Crop Plants under Drought Stress"

_plants, 2021, doi:10.3390/plants10081730_

Round 1

Reviewer 1 Report

The review is enclosed as an attachment  below

Author Response

Manuscript number: Plants-1311896

Title: Green synthesized metal oxide nanoparticles mediated regulation of growth and physiology in crop plants under drought stress

Authors: N.M. Alabdallach, Md.M. Hasan, I. Hammamia, D. Alshehri, H.A. Alatawi

This is a review article focused on the green synthesis of titanium dioxide (TiO2), zinc oxide (ZnO) and iron oxide (Fe3O4) nanoparticles, their features, dosages and effect on plant growth and development under drought stress.

General remarks

In general, the review is clear and well written. However, there is a number of drawbacks that should be eliminated.

The title is clumsy constructed and should be rephrased. E.g., it may be “Green synthesized metal oxide nanoparticles mediate growth regulation and physiology of crop plants under drought stress” or “Mediating growth regulation and physiology of crop plants under drought stress by green synthesized metal oxide nanoparticles” etc.

Response: Thank you for your suggestions. The title has been revised and rephrased based on suggested title. The current title now “Green synthesized metal oxide nanoparticles mediate growth regulation and physiology of crop plants under drought stress”

In the Abstract (l. 20) authors refer to foliar application method as a basis of MONPs role in plants. However, the advantage of this method of application has not been particularly discussed in the article. If just this method of application was the subject of this review, it should have been mentioned in the title. Also, the authors stated that (l.25-26) “this review will contribute to our present understanding of MONPs outcome and activity in plants, as well as their uptake, transport, and functions in alleviating drought stress in plants”. In fact, there is no either information on the uptake and transport of MONPs in plant, or on the specific mechanism of action, but rather the effects of NONPs application on plant physiology are presented and discussed.

Response: Thank you for your comments. The coincidence of the title, abstract and other sections have been revised. The title has been rephrased according to your suggestions. The line (l.25-26) has been revised and rephrased. The word “uptake, transport, and functions” of MONPs have been deleted from the abstract section. Overall, the whole part of the abstract has been revised to ensure that it is consistent with the other sections of the paper.

In Keywords, their order should present the major points of interest, thus “nanoparticles” should be shifted to the beginning.

Response: Ok, Thank you. Nanoparticles has been written at the beginning of the key words.

The sentence in l. 83-85 is clumsy and hardly understandable, and should be rephrased.

Response: Thank you. This sentence has been revised and rephrased. Please check the line now.

The authors comprehensively presented the green synthesis of MONPs and the effects of MONPs application on the plant physiology and growth under drought stress in two figures and two tables. In the text of sections 2.1, 2.2 and 2.3, however, there is too many repetitions of what is already shown in Table 1. It would be better to make a more order in Table 1 by ranking not only by substrate e.g. fungi, bacteria and plants, but also by increasing size/scope within each substrate, e.g. for TiO2 , Bacteria, Aeromonas hydrophila, 28-54; Bacillus mycoides 40-60, Lactobacillus sp. 10-70; Plants, Cicer aeristinum L., 14; Annona squamosal L., 23; Solanum trilobatum L., 70, Jatropha curcas L., 25-100; Moringa oleifera Lam., 100 etc.;

Response: Thank you for bringing this comment. In table 1, we have followed your suggestions. The complete table rearranged based on substrate and nanoparticle size rank. Please check the table 1.

“nm” should not be repeated throughout the column Size, as in is already given in the head.

Response: Thank you. “nm” has been deleted from the column size as it is already given in the head. Please check it.

In Table 1, more biological sources may be listed than discussed in the text, but not conversely, as it is with respect to Fe2O3. Text is considered as a source of additional, more detailed information on what is listed in Table. In the sections 2.1, 2.2, 2.3, only other relevant information than that already given in Table 1 should be provided.

Response: Thank you for bringing this point. We have discussed and added new lines in the section 2.1,2.2., and 2.3. that already given in the Table 1. Please check section 2.1, 2.2, 2.3.

In Table 2, the effects of exogenous application of MONPs on drought stress tolerance is presented. This is an informative table. However, there are some unclear points, in particular in the “Metal oxide treatments” column. The treatments are given in % concentrations and in mg L-1, but also in mg kg-1 and in ppm that are also mg kg-1 (these units as non-SI units should not be used in the article, but converted to mg kg-1). While % concentrations and mg L-1 clearly suggest foliar application, in the case of mg kg-1 this shows application to soil. If so, two different methods of MONP application have been mixed in the table entitled “Exogenous application”. This needs to be clarified, and the methods distinguished and commented.

Response: Thanks for your valuable suggestions. In table 2, Metal oxide treatments” units have been converted to SI.

In the line 3 of Table 2, “Outcome” column, “Increased negative effects” should be probably “Decreased negative effects”.

Response: Thank you. Yes, it should be ‘decreased negative effects” and we have revised and corrected it.

In column “Metal oxide treatments” all units should be converted to SI and their writing unified (e.g. mg L-1 or mg/L, and not both, or mg l-1).

Response:Metal oxide treatment” units have been converted to SI unit as mmol/L and it was written as unified unit.

There is no discussion, why the MONPs are not yet widely applied in agriculture and what issues have to be resolved (costs, unknown long-term health/environmental effects, other issues??) This should be added, and at least mentioned.

Response: Thank you. We have mentioned and added some new lines in the introduction section. Please check the introduction section.

The language is, in general, correct, however needs through editing due to many typos, sometimes wrong indefinite/definite article use, small incorrections (e.g. l. 30 “to grow almost 9.6 billion..”, l. 56 “for an instance”, l. 50 “increase a plants resistance..” l.70-71 “that involve the use ….that require a large”, l. 240-241”..drought stress on photosynthetic activity, increasing photosynthetic activity ….”, missed spaces e.g. l. 264 “53nm”, incorrect units etc.

Response: Thank you. All minor corrections have been added in manuscript. Text now reads: to reach nearly 9.6 billion, For an example, increase plant tolerance, “that involve the use ….which requires a large”,beneficial effect under drought stress. These nanoparticles increased…”. “53 nm given spaces”

To conclude:

The article needs some moderate, but detailed and attentive revision.

Response: Thank you for your comments. We have extensively revised based on your suggestions.

Reviewer 2 Report

In this paper, Alabdallah and co-workers have presented a review on how green synthesized metal oxide nanoparticles can mediate regulation of growth and physiology in crop plants under drought stress. The paper is interesting; however, some substantial improvements are needed prior to publishing.

Major remarks

  1. Synthesis of metal oxide nanoparticle

This section should be supplemented with information about methods that are used to evaluate the size, quality and the concentration of green synthesized MONPs.

  1. Factors affecting the metal oxide nanoparticles

This section is poorly written. The major remark that is that it deals with the conditions applied for synthesis of nanoparticles in general. Majority of the references cited in this section are for the synthesis of silver or gold nanoparticles which are not the subject of this manuscript! Authors should put some effort to find papers dealing with how mentioned parameters affect the green synthesis of MONPs by plants! Alternatively, they can omit this section!

  1. Mode of action of metal oxide (MONPs) nanoparticles under drought stress

In this section authors give an overview of the effects produced by application of MONPs on plant exposed to drought stress, but they do not give any mechanism by which MNOPs can alleviate stressful conditions and promote plant growth. Mechanism should also be included.

Moreover, they only give examples of positive effects although in the Table 2 they presented a study in which TiO2NPs increased negative effects of drought on what plants. Negative aspects of MONPs application should also be commented!

The English language is not of the very good quality and professional help should be acquired for language editing. The manuscript should be checked for language by either English native speaker or by professional English language service.

All other specific remarks can be found in the pdf of the manuscript.

Author Response

In this paper, Alabdallah and co-workers have presented a review on how green synthesized metal oxide nanoparticles can mediate regulation of growth and physiology in crop plants under drought stress. The paper is interesting; however, some substantial improvements are needed prior to publishing.

Response: Thank you very much for your inspiring comments.

Major remarks

Synthesis of metal oxide nanoparticle

This section should be supplemented with information about methods that are used to evaluate the size, quality and the concentration of green synthesized MONPs.

Response: Thank you very much for comments. We have revised the whole section based on your comments and reviewer # 1 comments. We have supplemented with information about methods in the section#2 that are used to evaluate the size, quality and the concentration of green synthesized MONPs. Moreover, we have added more biological substrate and sources with methods. Please see the section #2.

Factors affecting the metal oxide nanoparticles

This section is poorly written. The major remark that is that it deals with the conditions applied for synthesis of nanoparticles in general. Majority of the references cited in this section are for the synthesis of silver or gold nanoparticles which are not the subject of this manuscript! Authors should put some effort to find papers dealing with how mentioned parameters affect the green synthesis of MONPs by plants! Alternatively, they can omit this section!

Response: Thank you. Based on your suggestions, alternatively, we agree to omit this section.

Mode of action of metal oxide (MONPs) nanoparticles under drought stress

In this section authors give an overview of the effects produced by application of MONPs on plant exposed to drought stress, but they do not give any mechanism by which MNOPs can alleviate stressful conditions and promote plant growth. Mechanism should also be included.

Response: Thank you for your comments. We have summarized the positive effects of MONPs on plant along with underlying general mechanisms exposed to drought stress. We have shown the general proposed mechanisms, which are shown on figure 2. In addition, some new lines have been added in favor of underlying mechanisms, particularly describing the antioxidant enzymes, and phenolic compounds. Please check section#3.

Moreover, they only give examples of positive effects although in the Table 2 they presented a study in which TiO2 NPs increased negative effects of drought on what plants. Negative aspects of MONPs application should also be commented!

Response: Yes, the Table 2 showed the positive effects of MONPs under drought stress. However, it is minor typing mistakes regarding negative effects, therefore it would be the “ TiO2NPs decreased negative effects of drought”. Please check the table now.

The English language is not of the very good quality and professional help should be acquired for language editing. The manuscript should be checked for language by either English native speaker or by professional English language service.

Response: The English language have been checked and edited by the native speaker.

All other specific remarks can be found in the pdf of the manuscript.

Response: Thank you. All other specific remarks have been addressed in the original text file.